# Cardiometabolic risk factors in social housing residents: A multi-site cross-sectional survey in older adults from Ontario, Canada

Gina Agarwal [1,2]*, Janice Lee[1,3], Homa Keshavarz[1], Ricardo Angeles[1], Melissa Pirrie[1], Francine Marzanek[1]

1 Department of Family Medicine, McMaster University, Hamilton, Ontario, Canada, 2 Department of Health Research Methods, Evidence, and Impact, McMaster University, Hamilton, Ontario, Canada, 3 Department of Medicine, McMaster University, Hamilton, Ontario, Canada

* gina.agarwal@gmail.com

## Abstract

### Objective

This study describes cardiometabolic diseases and related risk factors in vulnerable older adults residing in social housing, aiming to inform primary care initiatives to reduce health inequities. Associations between sociodemographic variables, modifiable risk factors (clinical and behavioural), health-related quality of life and self-reported cardiometabolic diseases were investigated.

### Design, setting, and participants

This was a cross-sectional study with an interviewer-administered questionnaire. Data was collected from residents aged 55 years and older residing in 30 social housing apartment buildings in five regions in Ontario, Canada.

### Outcome measures

The proportion of cardiometabolic diseases and modifiable risk factors (e.g., clinical, behavioural, health status) in this population was calculated.

### Results

Questionnaires were completed with 1065 residents: mean age 72.4 years (SD = 8.87), 77.3% were female, 87.2% were white; 48.2% had less than high school education; 22.70% self-reported cardiovascular disease (CVD), 10.54% diabetes, 59.12% hypertension, 43.59% high cholesterol. These proportions were higher than the general population. Greater age was associated with overweight, high cholesterol, high blood pressure and CVD. Poor health-related quality of life was associated with self-reported CVD and diabetes.

**Data Availability Statement:** Data cannot be shared publicly because of privacy and ethics restrictions. It can be made available in aggregate format after consultation with the corresponding

author or McMaster University's Department of Family Medicine Information Technology Manager (adamzcy@mcmaster.ca) to ensure long-term data storage and availability.

**Funding:** This work was supported by the Hamilton Academic Health Sciences Organization and Canadian Institutes of Health Research under Grant #133563. The funders had no role in study design, data collection and analysis, decision to publish, or preparation of the manuscript.

**Competing interests:** The authors have declared that no competing interests exist.

## Conclusions

Older adults residing in social housing in Ontario have higher proportion of cardiovascular disease and modifiable risk factors compared to the general population. This vulnerable population should be considered at high risk of cardiometabolic disease. Primary care interventions appropriate for this population should be implemented to reduce individual and societal burdens of cardiometabolic disease.

## Introduction

Older adults with low-income levels are a vulnerable population with poorer health status compared to the general population due to higher rates of chronic disease, advancing age and poverty [1]. Independently, these are all risk factors for poorer health status, but they also act synergistically. For instance, older adults have higher rates of chronic disease relative to their younger counterparts. Canadian statistics indicate that the prevalence of chronic disease in ages 65–79 is 42.7%, and in those aged more than 80, is 51.5% compared to the total population prevalence average of 29.2% [2,3]. Similarly, recently housed individuals who are vulnerable have a high prevalence of chronic disease; based on estimates from three major Canadian urban centers the prevalence was as high as 85% [4].

Seniors comprise a large proportion of the social housing population and by default have a low-income. The Ontario Non-Profit Housing Association reports that one third of individuals waiting for social housing are over age 65 [2,5,6]. These statistics have major implications for future health care provision, as Canada's population is projected to age rapidly, and the demand for subsidized housing is increasing, with 171,360 households already on the waiting list as of 2016 [2,3,5,6]. Consequently, it is important to study the patterns of disease in vulnerable older adults living in social housing, who may have a higher burden of chronic disease. Though this population has been shown to have health care providers [7] there are some challenges with accessing the healthcare in a timely fashion [8]. Currently, there is limited evidence on the health status and healthcare needs of older adults living in social housing. Due to low literacy rates and research participation levels in low-income social housing residents, traditional population surveys are ineffectual [9]. However, understanding this growing population's health status and needs are crucial to inform healthcare system resource allocation and efficiency.

We hypothesize that cardiometabolic diseases and their modifiable clinical risk factors (e.g. high blood pressure, high cholesterol, overweight) and modifiable behavioural risk factors (e.g. smoking) exist in greater proportions in older adults residing in social housing. This is consistent with previous research showing that hypertension and diabetes are more common in seniors generally but with higher rates in those of low socioeconomic status [10,11]. Understanding the rates of cardiometabolic risk factors in older adults living in social housing can help optimize future cardiometabolic screening and preventative interventions. Key risk factors included were behavioural (i.e., tobacco/smoking, alcohol, physical activity, diet including fatty food and fruits and vegetables consumption) and cardiometabolic diseases (i.e., overweight/obese, hypertension, diabetes, and elevated serum cholesterol). Key socio-demographic factors and known influencers of cardiometabolic risk, such as age, gender, education level, marital status and ethnicity were also considered.

In order to inform primary care initiatives to reduce health inequities, the objectives of this study were to report, for this vulnerable population of older adults residing in social housing, the:

1. proportions of cardiometabolic risk factors,

2. associations between the modifiable clinical risk factors and social determinants of health,

3. associations between cardiovascular diseases (CVD, which refers to congestive heart failure, hypertension, hypertension, atrial fibrillation, myocardial infarction and cerebrovascular accident) and diabetes, and social determinants of health,

4. associations between CVD and diabetes, and modifiable behavioural risk factors.

## Methods

### Study design

This was a cross-sectional analytical study using baseline data from a multi-site randomized controlled trial (RCT) of a community health intervention in social housing conducted between January 2015 and November 2016 [12]. These residential units were typically government subsidized at 30% of household income (i.e. rent-geared-to-income social housing). Participants were surveyed to collect a variety of measures described below.

### Participants

This study was conducted in thirty seniors' social housing apartment buildings in 5 urban and semi-urban centers across Ontario, Canada: Hamilton, Guelph-Wellington, York Region, Sudbury, and the County of Simcoe. The building selection criteria included: 1) building occupancy of 60% or more by individuals 55-years and older; 2) building size of 50 or more apartment units; 3) had a unique postal code; and 4) there was a similar building available (size, demographics, location, etc.). Seniors' units in social housing were typically reserved for those age 55 and older. Therefore, all residents 55 years old and older were included in the current study sample. Those living in the building for less than three months were excluded. Recruitment took place via consecutive sampling through posters in building common areas, with the advertisement of a ten-dollar gift card. Participants were surveyed individually by trained research staff through a structured survey. All surveys were conducted in English. Written informed consent was obtained using lay language at the Grade 6 level and explained verbally. Research ethics approval was obtained for the study from the Hamilton Integrated Research Ethics Board(#14–210 and #14–645).

### Measures

The Health Awareness and Behaviour Tool (HABiT) was used in this study to assess multiple domains of general health status [13]. It was specifically designed for use in older adults and has been previously piloted and validated with a sample of older adults in Ontario [13]. HABiT questions used in this study were obtained from validated surveys including the Canadian Hypertension Education Program (CHEP), (CHEP) [14], the Community Health Survey (CCHS) [15], the Canadian Diabetes Risk Questionnaire (CANRISK) [16], the EuroQual Five Dimensions (EQ5D-3L) [17], and the Newest Vital Sign UK (NVS-UK) [18].

Participants were assessed for their social determinants of health, cardiometabolic disease status, modifiable clinical and behavioural risk factors for cardiometabolic disease, self-reported health status, and health-related quality of life. Standard definitions for clinical, behavioural, and cardiometabolic diseases (CVD and diabetes) were followed. Social determinants of health included gender, age group, educational status, marital status, ethnicity, self-reported health status, and Health-Related Quality of Life (HRQoL), which included mobility, self care,

usual care, pain, and anxiety. Modifiable clinical risk factors were overweight (BMI >25kg/m$^2$), raised blood pressure (self-reported diagnosis or self-reported taking antihypertensive medications), and raised cholesterol (self-reported diagnosis or self-reported taking cholesterol lowering medications). Modifiable behavioural risk factors included fruits and vegetable consumption, physical inactivity, alcohol consumption, salt added to food, fatty food consumption, and smoking.

## Data analysis

Descriptive statistics were used to summarize responses to the HABiT questionnaire for all sociodemographic characteristics, self-reported cardiometabolic diseases, and related risk factors.

Significant associations between modifiable clinical and behavioural, health status and HRQoL risk factors and cardiometabolic diseases (CVD and diabetes) were determined using non parametric tests of association (Pearson chi-squared or Fisher exact test, as appropriate). Proportions (%) and 95% confidence intervals (CIs) were then calculated for each subgroup. Missing data ranged from 0.3 to 2.2%, except for BMI, which was 5%. Since BMI was calculated from weight and height, where these variables were missing, BMI was also coded with a "missing" category. Data were analyzed using IBM SPSS Statistics version 28.0 [19].

## Results

A total of 1065 participants were included in this cross-sectional analysis. The mean age of the sample was 72.4 years (standard deviation (SD) = 8.87); 77.3% were female; 87.2% were White, 6.2% were Indigenous and 6.7% were other ethnicities. Participants typically had low education levels; 48.2% of participants had some high school and 21.0% had completed high school (Table 1).

The distributions of three self-reported modifiable clinical risk factors (elevated blood pressure (BP), raised cholesterol, and overweight/obese) were less heterogeneous compared to modifiable behavioural risk factors. A higher proportion of respondents in the over 85-year age group reported hypertension compared to those in the 55-to-64-year age group (60.6% vs 48.6%). However, a smaller proportion of respondents in the over 85-year age group reported high cholesterol compared to the 55-to-64-year age group (30.9% vs 41.9%) and those overweight (48.9% vs 71.3%). In terms of ethnicity, participants who identified as Indigenous had the highest proportion of reported overweight (72.7%) compared to 70.5% for those who were white). No differences in modifiable clinical risk factors were observed based on sex, marital status, and education (Table 2).

The proportion of reported CVD was higher among the 75–84 age group compared to younger age groups and the over 85 years age group (Table 3). The proportion of individuals with diabetes was higher amongst those who lived alone (25.1% to 33.9%) as opposed to being married (18.7%) (Table 3).

Among female participants, the proportion that reported having diabetes was highest among non-drinkers (30.3%) compared to low-risk drinkers (19.8%) and high-risk drinkers (12.5%). A higher proportion of overweight individuals had diabetes (34% compared to 15.4% of non-overweight individuals) (Table 4).

Over one third of individuals with CVD (33.9%) and with diabetes (39.5%) indicated their self-reported health status as poor. In terms of mobility, 29.4% of individuals with CVD and 35.3% of individuals with diabetes indicated having extreme mobility problems. A significantly higher proportion of individuals with CVD indicated having extreme self care problems (45.2%) compared with those who indicated no problems. Similarly, a significantly higher

**Table 1. Proportion of sociodemographic determinants, clinical risk factors, and cardiometabolic diseases.**

| Variables | | Number (%)/ Mode/Mean ± SD |
|---|---|---|
| **Sex** | Male | 242 (22.72%) |
| | Female | 823 (77.28%) |
| | Total | 1065 |
| **Age (years)** | 55–64 | 261(24.51%) |
| | 65–74 | 405 (38.03%) |
| | 75–84 | 304 ((28.54%) |
| | ≥85 | 95 (8.92%) |
| | **Total** (Mean ± SD) | 1065 (72.36±8.8) |
| **Education** | Some High School | 504 (48.23%) |
| | High School Diploma | 219 (20.96%) |
| | Some College/University | 155 (14.83%) |
| | College/University Degree | 167 (15.98%) |
| | **Total** (Mode) | 1045 (Some High School) |
| **Marital Status** | Married | 93 (8.84%)) |
| | Separated | 107 (10.17%) |
| | Divorced | 380 (36.12%) |
| | Widowed | 332 (31.56%) |
| | Never Married | 140 (13.31%) |
| | Total (Mode) | 1052 (Divorced) |
| **Ethnicity** | White | 882 (87.15%) |
| | Indigenous | 63 (6.23%) |
| | Others* | 67 (6.62%) |
| | Total (Mode) | 1012 (White) |
| **Clinical Risk Factors** | High Blood Pressure | 619 (59.12%) |
| | High Cholesterol | 454 (43.59%) |
| | Overweight | 704 (69.29%) |
| **Cardiometabolic Diseases** | CVD | 239 (22.70%) |
| | Diabetes | 301 (28.59%) |

*Other non-white (Latin American, Arab, West Asian)

CVD: cardiovascular disease

proportion of individuals with CVD indicated having extreme usual care problems (41.3%) compared with indicating no problems. In terms of pain, individuals with CVD were more likely to indicate having extreme problems (32.4%) compared to no problems (15.9%). Individuals with diabetes were also more likely to indicate extreme problems in terms of usual care (41.3%) and pain (35.8%) (Table 5).

## Discussion

There are few studies examining modifiable cardiometabolic risk factors among low-income older adults residing in social housing in Ontario. In the current study we explored social determinants of health, modifiable clinical and modifiable behavioural risk factors, self-reported health status, and health-related Quality of Life determinants and their associations with cardiometabolic diseases (CVD and diabetes). Overall, our findings indicate that regardless of their CVD or diabetes status, social housing participants reported poor health behaviours, quality of life, and general health status. A significantly higher proportion of individuals with CVD indicated having extreme problems with self care, usual care, and pain. Similarly,

**Table 2. Proportion of self-reported modifiable clinical risk factors stratified by sociodemographic determinants.**

| Sociodemographic determinants | | Modifiable Clinical RF | | |
|---|---|---|---|---|
| | | High Blood Pressure | High Cholesterol | Overweight* |
| | | Proportion (95% CI) | Proportion (95% CI) | Proportion (95% CI) |
| **Sex** | Male | 59.14 (52.73 to 65.19) | 44.1 (40.7 to 47.5) | 65.2 (59.0 to 71.1) |
| | Female | 59.07 (55.72 to 62.48) | 41.8 (35.7 to 48.2) | 70.5 (67.2 to 73.6) |
| | *P-value* | p = 0.986 | p = 0.535 | *p* = 0.128 |
| **Age (years)** | 55–64 | 48.6 (42.6 to 54.7) | 41.9 (36.06 to 47.9) | 71.3 (65.5 to 76.6) |
| | 65–74 | 60.8 (55.96 to 65.4) | 47.4 (42.56 to 52.3) | 73.5 (68.9 to 77.7) |
| | 75–84 | 65.5 (60.0 to 70.8) | 44.0 (38.5 to 49.7) | 68.2 (62.6 to 73.3) |
| | ≥85 | 60.6 (50.6 to 70.1) | 30.9 (22.2 to 40.7) | 48.9 (36.6 to 59.2) |
| | *P-value* | **p = 0.001** | **p = 0.031** | **p<0.001** |
| **Education** | Some High School | 58.1 (53.7 to 62.3) | 44.0 (39.8 to 48.4) | 66.0 (61.7 to 70.1) |
| | High School Diploma | 60.6 (54.0 to 66.0) | 38.5 (32.3 to 45.1) | 73.7 (67.5 to 79.3) |
| | Some College/University | 61.3 (53.5 to 68.7) | 47.1 (39.4 to 54.9) | 75.3 (68.0 to 81.7) |
| | College/University Degree | 57.5 (49.9 to 64.8) | 44.3 (36.9 to 51.9) | 67.3 (59.9 to 74.1) |
| | *P-value* | p = 0.826 | p = 0.375 | *p* = 0.062 |
| **Marital Status** | Married | 60.2 (49.8 to 7.00) | 43.3 (33.4 to 53.6) | 70.5 (60.4 to 92.5) |
| | Separated | 57.5 (48.0 to 66.6) | 56.6 (47.4 to 65.8) | 65.4 (55.6 to 70.4) |
| | Divorced | 57.4 (52.4 to 62.3) | 43.5 (38.6 to 48.6) | 69.3 (64.4 to 73.8) |
| | Widowed | 63.8 (58.5 to 68.9) | 41.8 (36.6 to 47.2) | 69.9 (64.7 to 74.7) |
| | Never Married | 52.2 (43.9 to 60.4) | 46.4 (38.3 to 54.7) | 70.0 (61.7 to 77.4) |
| | *P-value* | p = 0.173 | p = 0.930 | p = 0.926 |
| **Ethnicity** | White | 59.3 (56.0 to 62.5) | 43.8 (40.5 to 47.1) | 70.5 (67.4 to 73.5) |
| | Indigenous | 56.1 (43.2 to 68.5) | 45.6 (33.2 to 48.5) | 72.7 (60.0 to 83.1) |
| | Other* | 65.1 (52.8 to 76.0) | 46.9 (35.0 to 59.0) | 43.5 (32.2 to 55.2) |
| | *P-value* | p = 0.575 | p = 0.847 | **p = 0.023** |

*Other: non-white (Latin American, Arab, West Asian)

individuals with diabetes were significantly more likely to indicate extreme problems with usual care and with pain.

Overall, these results are consistent with previous studies [9,23–25]. Compared with the average Canadian of a similar age, residents in social housing consumed fewer fruits and vegetables per day. The Canadian Community Health Survey (CCHS) suggests that less than 5% of Canadians over age 65 consume less than one serving of fruits and vegetables per day [26]. Comparatively, one third of participants in our study failed to consume fruits and vegetables even one time per day. The proportion of individuals who had CVD and added salt "sometimes" or "always" to their food was high at 38.7% and 31.9%, respectively. Although there is a lack of Canadian data for this exact measures, national statistics on sodium consumption suggest that of those aged 51–70 years, 70% of males and 31% of females consume excessive sodium, and of those aged 71 years or older, 56% of males and 24% of females consume excessive sodium (which was defined as 2300 mg per day) [15]. The CCHS data only reports that the average Canadian consumes 146 kcal per day from fast food [15]. Similarly, rates of tobacco consumption and alcohol consumption in males in our study, were higher than the general Canadian population of the same age. However, the proportion of participants who exercised for at least 30 minutes per day is similar to 40.6% of Canadians over age 65, who exercised more than 150 minutes per week also reported in the National data [15].

**Table 3. Proportion of self-reported cardiometabolic diseases (CVD and diabetes) stratified by sociodemographic determinants.**

| Sociodemographic determinants | | Cardiometabolic Diseases | |
|---|---|---|---|
| | | **CVD** | **Diabetes** |
| | | **Proportion (95% CI)** | **Proportion (95% CI)** |
| **Sex** | Male | 22.2 (19.5 to 25.2) | 29.6 (24.1 to 35.6) |
| | Female | 24.3 (19.2 to 30.0) | 28.3 (25.2 to 31.4) |
| | p-value | p = 0.510 | p = 0.689 |
| **Age** | 55–64 | 19.4 (14.9 to 24.5) | 30.6 (25.2 to 36.4) |
| | 65–74 | 19.5 (15.8 to 23.5) | 29.7 (25.4 to 34.3) |
| | 75–84 | 29.0 (24.1 to 34.9) | 28.2 (23.4 to 33.5) |
| | ≥85 | 25.5 (17.6 to 35.0) | 19.1 (12.2 to 28.0) |
| | p-value | p = 0.010 | p = 0.181 |
| **Education** | Some High School | 21.8 (18.4 to 31.8) | 29.0 (25.1 to 33.0) |
| | High School Diploma | 25.7 (20.2 to 30.4) | 28.0 (22.3 to 34.2) |
| | Some college/university | 26.5 (20.0 to 33.8) | 27.1 (20.6 to 34.5) |
| | College/university degree | 24.0 (18.0 to 30.8) | 29.3 (22.8 to 36.6) |
| | p-value | p = 0.587 | p = 0.962 |
| **Marital Status** | Married | 17.8 (11.0 to 26.6) | 18.7 (11.7 to 27.6) |
| | Separated | 18.9 (12.3 to 27.1) | 30.2 (22.1 to 39.4) |
| | Divorced | 21.9 (18.0 to 26.3) | 25.1 (20.9 to 29.6) |
| | Widowed | 27.9 (23.2 to 32.9) | 33.9 (29.0 to 39.2) |
| | Never Married | 19.3 (13.4 to 26.4) | 29.3 (22.2–37.2) |
| | p-value | p = 0.084 | p = 0.021 |
| **Ethnicity** | White | 23.9 (21.2 to 26.8) | 27.8 (24.9 to 30.8) |
| | Indigenous | 12.3 (5.7 to 22.6) | 39.3 (27.3 to 52.4) |
| | Other* | 50.0 (24.3 to 75.7) | 32.8 (22.3 to 44.9) |
| | p-value | p = 0.308 | p = 0.392 |

Notes: p-values reported here are from Chi squared and Fisher's Exact tests; *Other: non-white (Latin American, Arab, West Asian)

Reasons that populations in social housing have high rates of behavioural risk factors are many. Systematic barriers faced by individuals living in poverty may be important factors. For instance, social housing participants may lack access to healthy food choices due to geography, income, poor mobility, and social isolation [27]. Neighborhoods with lower socioeconomic status are less likely to have grocery stores or healthier fast food options [28,29]. Secondly, the few fruits and vegetables offered in areas with poor grocery store access often have higher prices [30]. Thirdly, our participants had a high proportion of pain and mobility issues, which can directly limit their ability to physically access healthy food on a regular basis. Lastly, poverty and poor health literacy may also represent potential multidirectional mechanisms for higher rates of smoking and alcohol consumption [31]. Likewise, participants had poorer health status than the average Canadian within a similar age range. Diabetes and hypertension rates were higher in our population than the national average for people aged 65 and older (28.6% vs. 17.9%; 59.1% vs. 43.7%) [15]. Similarly, a larger proportion of social housing participants were either overweight or obese (69.3% of all participants and especially people with diabetes at 79.4%). Compared to the age—matched national average (68.3%), people with diabetes in our study were more likely to be overweight or obese [15]. In terms of quality of life, social housing participants reported higher proportion of pain and mobility issues (69.7–74.1% vs. 51%; 57.9–67.9% vs. 22%).%) [17]. These findings are consistent with higher rates of chronic medical diseases, and comorbidities in social housing residents and those with lower

**Table 4. Proportion of cardiometabolic disease (CVD and DM) along with its 95% confidence interval stratified by modifiable behavioural risk factors.**

| Modifiable behavioural risk factors | | Cardiometabolic disease | |
| --- | --- | --- | --- |
| | | CVD | Diabetes |
| | | Proportion (95% CI) | Proportion (95% CI) |
| **Fruits and Vegetables** | Not everyday | 20.7 (17.0 to 24.8) | 25.6 (21.5 to 30.0) |
| | Everyday | 24.0 (20.8 to 27.4) | 30.2 (26.7 to 33.8) |
| | *P-value* | *p = 0.219* | *p = 0.106* |
| **Physical Activity** | 30 minutes per day | 23.0(19.6 to 26.7) | 26.4 (22.8 to 30.3) |
| | Less than 30 per day | 22.4 (19.0 to 26.1) | 30.5 (26.6 to 34.5) |
| | *P-value* | *p = 0.820* | *p = 0.143* |
| **Alcohol drinking/ week (Female)** | Non—drinker | 22.5 (19.8 to 26.2) | 30.3 (26.9–33.8) |
| | Low-risk drinking* | 19.1 (13.1 to 26.4) | 19.8 (13.7 to 27.3) |
| | High-risk drinking** | 18.8 (5.6 to 42.1) | 12.5 (2.7 to 34.4) |
| | *P-value* | *p = 0.597* | **p = 0.020** |
| **Alcohol drinking/ week (Male)** | Non—drinker | 23.1 (17.1 to 30.1) | 29.4 (22.7 to 36.8) |
| | Low-risk drinking* | 25.7 (16.6 to 36.8) | 28.6 (19.0 to 39.9) |
| | High-risk drinking** | 25.0 (5.6 to 59.2) | 25.0 (5.6 to 59.2) |
| | *P-value* | *p = 0.912* | *p = 0.961* |
| **Salt Added to food** | Always | 31.9 (20.0 to 46.0) | 34.0 (21.8 to 48.2) |
| | Often | 17.9 (7.2 to 34.8) | 21.4 (9.5 to 38.9) |
| | Sometimes | 38.7 (27.3 to 51.1) | 27.4 (17.5 to 39.4) |
| | Rarely | 22.4 (14.1 to 32.7) | 27.6 (18.5 to 38.4) |
| | Never | 23.9 (16.8 to 32.3) | 30.1 (22.2 to 39.0) |
| | *P-value* | *p = 0.114* | *p = 0.814* |
| **Fatty Food Consumption per week** | Everyday | 20.0 (11.8 to 32.4) | 31.0 (20.3 to 43.6) |
| | 3–4 meals/week | 15.3 (10.7 to 23.9) | 25.6 (18.5 to 33.9) |
| | 1–2 meals/week | 24.0 (19.8 to 28.7) | 28.9 (24.3 to 33.8) |
| | 2–3 meals/month | 26.0 (20.0 to 32.8) | 29.4 (23.0 to 36.4) |
| | Never | 22.3 (18.2 to 26.9) | 28.4 (23.8 to 33.3) |
| | *P-value* | *p = 0.378* | *p = 0.943* |
| **Smoking** | No | 22.4 (19.5 to 25.5) | 28.1 (24.9 to 31.4) |
| | Yes | 23.1 (18.7 to 28.1) | 29.3 (24.4 to 34.6) |
| | *P-value* | *p = 0.798* | *p = 0.684* |
| **Overweight** | No | 19.6 (15.5 to 24.3) | 15.4 (11.7 to 19.8) |
| | Yes | 24.3 (21.2 to 27.5) | 34.0 (30.6 to 37.6) |
| | *P-value* | *p = 0.105* | *p<0.001* |

Notes: p-values reported here are from Chi squared and Fisher's Exact tests; CVD = cardiovascular disease

*Low drinkers: Those considered low risk have intake as per the 2012 Canadian Guidelines (<10/week for women, <15/week for men or <5 drinks at once for either men or women).

**High drinkers: Those considered as having high weekly intake or binge drinkers by the 2012 Canadian Guidelines (>10/week for women, >15/week for men or >5 drinks at once more than once a month for either men or women) [20–22].

socioeconomic status [32]. Underlying reasons may include decreased access to primary and ambulatory care in low—income populations [32–37]. Another possible reason underlying our findings is that this population may be at risk because they have markedly lower education levels. According to the 2021 Census, Ontario seniors (65 years and older) generally have high educational attainment with 47.9% having completed a post-secondary education, 28.3% a high school diploma, and only 23.8% have not graduated from high school [38]; however, 48.2% of seniors in this study had not completed high school. This is consistent with literature

**Table 5. Proportion of self—reported cardiometabolic diseases stratified by self—reported health status and Health-Related Quality of Life.**

| Health Status & Health-Related Quality of Life | | Cardiometabolic Diseases | |
| --- | --- | --- | --- |
| | | CVD Proportion (95% CI) | Diabetes Proportion (95% CI) |
| **Self-Reported Health Status** | Excellent | 9.4 (4.0 to 18.3) | 10.9 (5.0 to 20.3) |
| | Very good | 11.9 (7.9 to 17.0) | 22.3 (16.8 to 28.5) |
| | Good | 19.3 (15.5 to 23.5) | 28.4 (24.0 to 33.1) |
| | Fair | 32.4 (27.3–37.9) | 31.7 (26.6 to 37.2) |
| | Poor | 33.9 (26.0–42.5) | 39.5 (31.2 to 48.3) |
| | *P-value* | *P* = 0.001 | *P* = 0.001 |
| **Health—Related Quality of Life** | | | |
| **Mobility** | No problems | 14.0 (11.1 to 17.6) | 21.6 (17.9 to 25.6) |
| | Some problems | 28.6 (25.1 to 32.4) | 33.3 (29.5 to 37.2) |
| | Extreme problems | 29.4 (12.2 to 53.0) | 35.3 (16.3 to 58.9) |
| | *P-value* | *P* = 0.001 | *P* = 0.001 |
| **Self Care** | No problems | 18.2 (15.7 to 20.9) | 26.4 (23.5 to 29.5) |
| | Some problems | 38.5 (31.6 to 45.8) | 35.2 (28.5 to 42.4) |
| | Extreme problems | 45.2 (28.7 to 62.5) | 41.9 (25.9 to 59.4) |
| | *P-value* | *P* = 0.001 | *P* = 0.014 |
| **Usual Care** | No problems | 15.7 (13.0 to 18.7) | 24.8 (21.5 to 28.3) |
| | Some problems | 31.5 (26.9 to 36.3) | 32.8 (28.2 to 37.7) |
| | Extreme problems | 41.3 (28.0 to 55.7) | 41.3 (28.0 to 55.7) |
| | *P-value* | *P* = 0.001 | *P* = 0.003 |
| **Pain** | No problems | 15.9 (12.2 to 20.2) | 23.8 (19.4 to 28.7) |
| | Some problems | 23.6 (20.3 to 27.2) | 28.8 (25.2 to 32.6) |
| | Extreme problems | 32.4 (25.3 to 40.3) | 35.8 (28.4 to 43.7) |
| | *P-value* | *P* = 0.001 | *P* = 0.026 |
| **Anxiety** | No problems | 19.5 (16.3 to 22.9) | 29.1 (25.4 to 33.0) |
| | Some problems | 24.8 (20.8 to 29.1) | 27.6 (23.5 to 32.0) |
| | Extreme problems | 35.6 (24.3 to 48.3) | 25.4 (15.7 to 37.5) |
| | *P-value* | *P* = 0.007 | *P* = 0.878 |

Notes: p-values reported here are from Chi squared and Fisher's Exact tests; CVD = cardiovascular disease

which has found both poorer knowledge of cardiometabolic disease risk factors and disparity in cardiac outcomes in lower-level socioeconomic groups [39,40].

Our results suggest that seniors living in social housing are a vulnerable group with poor cardiometabolic health. As such, there may be a role for cardiometabolic preventative public health programs, tailored toward this population that is distinct from those programs currently available. Previous studies have shown that interventions that target the entire community have lower rates of effectiveness for traditionally marginalized populations, and that if not targeted carefully, can exacerbate health disparities [41,42]. Thus, public health programs for this population must take participants' lack of financial resources, poor health literacy and poor mobility into account [9]. Lastly, specific interventions that aim to diagnose and better control hypertension are needed. These public health screening programs could target older age groups and those lacking in physical activity. In terms of program development for those already diagnosed with hypertension, interventions may present a more significant challenge. A multifaceted interventional program will be needed as other factors not evaluated in our study, such as lack of access to medication may also have a role in the observed associations.

Our study has several strengths. First, our study is one of the first to examine cardiometabolic risk factors in social housing residents using validated tools. This population is "hard to reach" with low response rates [9]. The total sample size of 1065 participants was relatively large. The study was conducted in diverse Ontario cities of varying sizes accounting for different environmental factors, all of which improve the generalizability of the findings to the larger Canadian population of adults residing in social housing. The limitations of the study include consecutive sampling, which may be affected by response bias. The study utilized self—selection, meaning that participants were possibly more health literate than the average senior living in social housing. Nonetheless our sample had remarkably lower education levels than the average population. Additionally, we collected self—reported health data, and thus, diagnoses of hypertension and diabetes, and self-reported health behaviours and weight/height used to calculate BMI, may be prone to errors. In addition, the majority of our study sample was White (87%), with a very small proportion of each non-white ethnicity represented in the remaining 13%. As such, generalizability of results to non-White populations is limited. The study utilized consecutive sampling to access the intended population who are underserved and understudied [9,23] and achieve the highest coverage possible. Initially, a random sampling design was used with a predefined list of unit numbers to be reached by door-knocking. However, this approach was not successful due to the general unwillingness of residents to open their doors to researchers, even with advanced notice. Consequently, the list of random unit numbers was continuously expanded until enough responders were reached and the required sample size was achieved, effectively becoming a consecutive sample. Therefore, consecutive sampling was ultimately used successfully from the outset for the remaining buildings. This may have increased the risk of response bias in our sample whereby the characteristics of responders may be expected to differ from non-responders.

## Conclusion

Overall, this study has described poor health behaviours, health status, and quality of life in social housing residents from thirty social housing buildings throughout Ontario. Poverty, poor health literacy, and social isolation may be potential mediators for these associations, but requires future research. Cardiometabolic preventative public health programs targeted toward health behaviours must take into account such factors as social and environmental circumstances to improve health outcomes in older adults living in social housing.

## Supporting information

**S1 Checklist. STROBE statement—checklist of items that should be included in reports of *cross-sectional studies*.**
(DOC)

## Author Contributions

**Conceptualization:** Gina Agarwal, Janice Lee, Homa Keshavarz, Ricardo Angeles, Melissa Pirrie, Francine Marzanek.

**Data curation:** Gina Agarwal, Janice Lee, Homa Keshavarz.

**Formal analysis:** Gina Agarwal, Homa Keshavarz, Ricardo Angeles.

**Investigation:** Gina Agarwal.

**Methodology:** Gina Agarwal, Homa Keshavarz, Ricardo Angeles.

**Supervision:** Gina Agarwal.

**Writing – original draft:** Gina Agarwal, Janice Lee, Homa Keshavarz, Ricardo Angeles.

**Writing – review & editing:** Gina Agarwal, Homa Keshavarz, Melissa Pirrie.

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
