## [Decision Letter · Decision Letter 0]

16 Jan 2024

PONE-D-23-35505Cardiometabolic risk factors in social housing residents: a multi-site cross-sectional survey in older adults from Ontario, CanadaPLOS ONE

Dear Dr. Agarwal,

Thank you for submitting your manuscript to PLOS ONE. After careful consideration, we feel that it has merit but does not fully meet PLOS ONE’s publication criteria as it currently stands. Therefore, we invite you to submit a revised version of the manuscript that addresses the points raised during the review process.

**ACADEMIC EDITOR: ** Thank you very much for submitting your manuscript to PLOS ONE. Three experts in the field have now assessed your paper. Please carefully address all reviewers' comments, especially the ones raised concerning the practical implications of the works. I would be happy to reconsider your work after the revisions. 

We look forward to receiving your revised manuscript.

Kind regards,

Amin Nakhostin-Ansari

Academic Editor

PLOS ONE

“This work was supported by the Hamilton Academic Health Sciences

Organization (HAHSO) and the Canadian Institutes of Health Research (CIHR) under Grant MOP-133563”

Reviewers' comments:

Reviewer's Responses to Questions

**Comments to the Author**

1. Is the manuscript technically sound, and do the data support the conclusions?

Reviewer #1: Yes

Reviewer #2: Yes

Reviewer #3: Yes

2. Has the statistical analysis been performed appropriately and rigorously? 

Reviewer #1: Yes

Reviewer #2: No

Reviewer #3: No

3. Have the authors made all data underlying the findings in their manuscript fully available?

Reviewer #1: Yes

Reviewer #2: No

Reviewer #3: Yes

4. Is the manuscript presented in an intelligible fashion and written in standard English?

Reviewer #1: Yes

Reviewer #2: Yes

Reviewer #3: Yes

5. Review Comments to the Author

Reviewer #1: I have had the opportunity to thoroughly review manuscript titled "Cardiometabolic risk factors in social housing residents: a multi-site cross-sectional survey in older adults from Ontario, Canada". In the following paragraphs, I provide an assessment of this work, evaluating its technical soundness, data analysis and presentation.

Technical Soundness: The manuscript demonstrates a high degree of technical soundness. The research methodology appears well planned and executed. The experimental design, data collection, and analysis are all appropriately documented. The study's objectives are clear, and the research questions are logically addressed. The use of appropriate methods and techniques in data collection and analysis enhances the manuscript's credibility.

Data and Conclusions: The data presented in the manuscript are comprehensive and well organized. It is evident that the data supports the conclusions drawn. The results align with the research objectives, and the conclusions are well founded. The logical flow of the manuscript aids in understanding the significance of the findings. This alignment between the data and the conclusions significantly strengthens the manuscript's overall quality.

Statistical Analysis: The statistical analysis in the manuscript is both appropriate and rigorous. The methods utilized for data analysis are well described, and the statistical tests chosen are relevant to the research questions. Moreover, the presentation of results in tables, figures, and the accompanying discussion contributes to the overall clarity of the statistical analysis.

Presentation and Clarity: The manuscript is well structured and presented in an intelligible fashion. The logical flow from introduction to methods, results, and discussion aids readers in following the research narrative. The tables are appropriately placed and add value to the text, helping to visualize the data and results.

In conclusion, manuscript is of high quality, both in terms of its technical content and presentation. The study is methodologically sound, the data analysis is robust, and the conclusions are well supported by the data.

Recommendations to the authors: Health related issues have been considered, however I recognized some missing information related to access of social housing residents to primary health care and medicines required to manage chronic conditions, including access to medicines used in diabetes, hypertension, etc. Such data will provide an additional strong evidence to policy decision makers in development of a specific program to address needs of social housing residents in health care

Reviewer #2: 1. Descriptive study of a local vulnerable population of social housing inhabitants in Ontaria Canada, whose generalizability and public health implications are lacking.

2. Line 148: How did you ascertain that BMI was expected to be missing not at random ?

3. Definition of CVD used in the current study should be clarified.

4. The analysis plan of objectives 2 – 4 in lines 91 – 93 is lacking.

Reviewer #3: 1. Although the paper's significance is understandable, the introduction can be paraphrased due to unnecessary repetition and needs better structure. Several times, you mentioned the modifiable and behavioral risk factors. Also, you repeatedly say the General purpose of the study ( lines 66-67, 71-72, and 95).

2. In Table 1, please add a percentage (%) mark in the related parenthesis to ease reading. Also, add a definition of the abbreviation in the footer of the tables ( CVD,…)

3. It would be better to describe the ranges( definition) of variables you asked for from participants in the method section. For example, the readers can only understand if they read the questionnaires separately) what low or high drinkers mean precisely.

4. Reports of health-related quality of life in the total study population( unless the cardiometabolic disease) can be informative, too. Add the entire population column in Table 5.

5. The descriptive reports of this article are important and support the purpose of the study well. On the other hand, analytical reports(chi-squares) often need to convey something meaningful to the reader, considering the purpose of the study. In fact, the purpose of the study, as evident in the discussion section by comparing the descriptive reports with the society by the authors, is that the reports of P values do not add anything new to the reader. The absence of most of them and only the descriptive report is enough for the study. In this study, we do not want to compare and correlate the effect of risk factors in diabetic or cardiac patients, but only their distribution in the targeted population is essential.

6. PLOS authors have the option to publish the peer review history of their article (what does this mean?). If published, this will include your full peer review and any attached files.

Reviewer #1: No

Reviewer #2: No

Reviewer #3: No

---

## [Author Response · Author response to Decision Letter 0]

26 Feb 2024

Please see the responses in the new cover letter/response to reviewers letter.

---

## [Decision Letter · Decision Letter 1]

8 Mar 2024

PONE-D-23-35505R1Cardiometabolic risk factors in social housing residents: a multi-site cross-sectional survey in older adults from Ontario, CanadaPLOS ONE

Dear Dr. Agarwal,

Thank you for submitting your manuscript to PLOS ONE. After careful consideration, we feel that it has merit but does not fully meet PLOS ONE’s publication criteria as it currently stands. Therefore, we invite you to submit a revised version of the manuscript that addresses the points raised during the review process.

**ACADEMIC EDITOR: ** Thank you very much for revising the manuscript according to the comments. The manuscript's quality has improved after the revisions. Please carefully address the comments by reviewer #3 and I will be happy to reconsider your work. 

We look forward to receiving your revised manuscript.

Kind regards,

Amin Nakhostin-Ansari

Academic Editor

PLOS ONE

Journal Requirements:

Reviewers' comments:

Reviewer's Responses to Questions

**Comments to the Author**

1. If the authors have adequately addressed your comments raised in a previous round of review and you feel that this manuscript is now acceptable for publication, you may indicate that here to bypass the “Comments to the Author” section, enter your conflict of interest statement in the “Confidential to Editor” section, and submit your "Accept" recommendation.

Reviewer #2: All comments have been addressed

Reviewer #3: All comments have been addressed

2. Is the manuscript technically sound, and do the data support the conclusions?

Reviewer #2: Partly

Reviewer #3: Yes

3. Has the statistical analysis been performed appropriately and rigorously? 

Reviewer #2: Yes

Reviewer #3: No

4. Have the authors made all data underlying the findings in their manuscript fully available?

Reviewer #2: Yes

Reviewer #3: Yes

5. Is the manuscript presented in an intelligible fashion and written in standard English?

Reviewer #2: Yes

Reviewer #3: Yes

6. Review Comments to the Author

Reviewer #2: All concerns were addressed by investigators and I have no new queries.

Therefore, no further comments.

Reviewer #3: Thanks to the authors for the interesting article and valuable revisions. The authors have made the necessary corrections precisely. However, as I mentioned in a comment, Reports of health-related quality of life in the total study population, unless the cardiometabolic disease, can be informative too. Although the authors wanted to compare the two major Cardiometabolic disorders groups, the reports of the total population as a vulnerable group can be informative, too.

There is no further comments

7. PLOS authors have the option to publish the peer review history of their article (what does this mean?). If published, this will include your full peer review and any attached files.

Reviewer #2: No

Reviewer #3: No

---

## [Author Response · Author response to Decision Letter 1]

15 Mar 2024

Thank you for the opportunity to review the paper again. We have uploaded the detailed response to each comment with the tracked changes of the original paper.

Comments Response

1. If the authors have adequately addressed your comments raised in a previous round of review and you feel that this manuscript is now acceptable for publication, you may indicate that here to bypass the “Comments to the Author” section, enter your conflict of interest statement in the “Confidential to Editor” section, and submit your "Accept" recommendation.

Reviewer #2: All comments have been addressed

Reviewer #3: All comments have been addressed

 Thank you for agreeing that the requirements have been met.

2. Is the manuscript technically sound, and do the data support the conclusions?

Reviewer #2: Partly

Reviewer #3: Yes

We believe based on the required comments that we have met the requirements requested of this point. 

3. Has the statistical analysis been performed appropriately and rigorously?

Reviewer #2: Yes

Reviewer #3: No

 We are aware that the 2 reviewers have differing opinions as to the statistical analysis that we performed, however we believe that this is common practice in research for there to be disagreement amongst academics over personal preference for styles of analysis performed. We are confident in our analytical techniques that we used, as several members of the research team have had extensive and rigorous statistical training at a doctorate level. Furthermore reviewer #3 commented in the previous review that inferential analysis was not required and that descriptive analysis was enough. Other reviewers might want to see the inferential analysis as well so we believe presenting both in this paper is the better option.

4. Have the authors made all data underlying the findings in their manuscript fully available?

Reviewer #2: Yes

Reviewer #3: Yes

Thank you for agreeing that the requirements have been met.

5. Is the manuscript presented in an intelligible fashion and written in standard English?

Reviewer #2: Yes

Reviewer #3: Yes

 Thank you for agreeing that the requirements have been met.

6. Review Comments to the Author

Reviewer #2: All concerns were addressed by investigators and I have no new queries.

Therefore, no further comments.

Reviewer #3: Thanks to the authors for the interesting article and valuable revisions. The authors have made the necessary corrections precisely. However, as I mentioned in a comment, Reports of health-related quality of life in the total study population, unless the cardiometabolic disease, can be informative too. Although the authors wanted to compare the two major Cardiometabolic disorders groups, the reports of the total population as a vulnerable group can be informative, too.

There is no further comments 

Thank you for agreeing that the requirements have been met.

Thank you reviewer #3 for pointing out that QoL can be informative - we were restricted by the space requirements of the tables and have therefore decided not to add another extra column in this case, which would make the table very cramped and hope you can accept this decision. We are preparing a separate paper that looks at QoL specifically in a larger social housing population in much more detail and do not feel it is relevant to place it here.

General Comment: 

Please review your reference list to ensure that it is complete and correct. If you have cited papers that have been retracted, please include the rationale for doing so in the manuscript text, or remove these references and replace them with relevant current references. Any changes to the reference list should be mentioned in the rebuttal letter that accompanies your revised manuscript. If you need to cite a retracted article, indicate the article’s retracted status in the References list and also include a citation and full reference for the retraction notice. Thank you for this - we have now checked the manuscript fully and made these 4 changes:

reference #3 substituted:

3. Canada PHAo. Center for Surveillance and Applied Research; 2017. substituted to

3. The 2017 Canadian chronic disease indicators SC. Health Promotion and Chronic Disease Prevention in Canada: Research, Policy and Practice. 2017;37(8):248.

reference #5 with minor changes updated:

5. Association ON-PH. Aging in place in social housing. 2016. Changed to

5. Ontario Non-Profit Housing Association (ONPHA). Aging in place in social housing Prepared by ONPHA 2016.

reference #15 with minor change updated:

15. Canada. S. Canadian Community Health Survey - Healthy Aging (CCHS) 2009. Changed to

15. Statistics Canada. Canadian Community Health Survey - Healthy Aging (CCHS) 2009.

reference #38 with minor change updated:

38. Canada. S. Highest level of education by geography:. Canada, provinces and territories. Changed to

38. Statistics Canada.Highest level of education by geography:. Canada, provinces and territories.

---

## [Editor Report · Decision Letter 2]

18 Mar 2024

Cardiometabolic risk factors in social housing residents: a multi-site cross-sectional survey in older adults from Ontario, Canada

PONE-D-23-35505R2

Dear Dr. Agarwal,

We’re pleased to inform you that your manuscript has been judged scientifically suitable for publication and will be formally accepted for publication once it meets all outstanding technical requirements.

Kind regards,

Amin Nakhostin-Ansari

Academic Editor

PLOS ONE
---

## [Editor Report · Acceptance letter]

26 Mar 2024

PONE-D-23-35505R2 

PLOS ONE

Dear Dr. Agarwal, 

I'm pleased to inform you that your manuscript has been deemed suitable for publication in PLOS ONE. Congratulations! Your manuscript is now being handed over to our production team.

Kind regards, 

on behalf of

Dr. Amin Nakhostin-Ansari 

Academic Editor

PLOS ONE